# Challenges in Drug and Hymenoptera Venom Hypersensitivity Diagnosis and Management in Mastocytosis

**DOI:** 10.3390/diagnostics14020123

**Published:** 2024-01-05

**Authors:** Matthew P. Giannetti, Jennifer Nicoloro-SantaBarbara, Grace Godwin, Julia Middlesworth, Andrew Espeland, Julia L. Douvas, Mariana C. Castells

**Affiliations:** 1Division of Allergy and Clinical Immunology, Brigham and Women’s Hospital, Boston, MA 02115, USA; 2Harvard Medical School, Boston, MA 02115, USA; 3Department of Psychiatry, Brigham and Women’s Hospital, Boston, MA 02115, USA

**Keywords:** mastocytosis, mast cell, nonsteroidal anti-inflammatory drugs (NSAIDs), antibiotics, vaccines, perioperative anaphylaxis

## Abstract

Mastocytosis is a myeloproliferative neoplasm characterized by abnormal proliferation and activation of clonal mast cells typically bearing the KITD816V mutation. Symptoms manifest due to the release of bioactive mediators and the tissue infiltration by neoplastic mast cells. Mast cell activation symptoms include flushing, pruritus, urticaria, abdominal cramping, diarrhea, wheezing, neuropsychiatric symptoms, and anaphylaxis. Up to 50% of patients with mastocytosis report a history of provoked and unprovoked anaphylaxis, with Hymenoptera venom and drugs the most common culprits. NSAIDs, antibiotics, vaccines, perioperative medications, and radiocontrast media are often empirically avoided without evidence of reactions, depriving patients of needed medications and placing them at risk for unfavorable outcomes. The purpose of this review is to highlight the most common agents responsible for adverse drug reactions in patients with mastocytosis, with a review of current epidemiology, diagnosis, and management of drug hypersensitivity and Hymenoptera venom allergy.

## 1. Introduction

Mastocytosis is a rare myeloproliferative neoplasm characterized by clonal proliferation, accumulation, and activation of mast cells (MCs) at multiple tissue sites such as the skin, bone marrow, and gastrointestinal (GI) tract [1,2,3]. Mastocytosis presents with multiple phenotypes, including cutaneous mastocytosis, systemic mastocytosis (SM), and MC sarcoma [4]. SM comprises non-advanced diseases such as indolent SM (ISM), the most frequent type of SM, and smoldering SM (SSM), and advanced diseases which include aggressive SM, MC leukemia (MCL), and SM with an associated hematologic neoplasm [5]. SM is estimated to affect 32,000 adults in the United States, with 95% of these cases categorized as nonadvanced [6]. Clonal mast cells bear KIT mutations, the most frequent being KITD816V, which has been recently targeted by new tyrosine kinase inhibitors aimed at modifying the natural history of the disease, inducing remissions and controlling symptoms of mast cell activation [7]. 

Patients with mastocytosis experience symptoms due to the increase in tissue mast cells and the acute and chronic release of membrane and granule mediators, and the generation of cytokines and chemokines. Granule mediators such as tryptase and histamine, and products of arachidonic acid such as prostaglandins and leukotrienes, bind to tissue receptors leading to a variety of symptoms including flushing, diarrhea, heartburn, dizziness, brain fog, headache, hypotension, and life-threatening anaphylaxis [8,9]. MC activation can be induced by specific and non-specific triggers [10] and commonly implicated triggers include drugs such as nonsteroidal anti-inflammatory drugs (NSAIDs), vaccines, opiates/opioids, neuromuscular blocking agents (NMBAs), radiocontrast media (RCMs), quinolones, vancomycin, and Hymenoptera venoms [11]. While anecdotal reports of reactions to these agents has been published, the true incidence and prevalence of drug reactions is unknown, and whether patients with cutaneous or systemic disease have increased risks has not been evaluated. Empiric avoidance is often recommended due to fear of potential reactions, placing patients at a disadvantage at the times of need, such as infection, surgery, and procedures or during the recent COVID-19 pandemic. Few efforts have been undertaken to de-label patients of inaccurate or unknown drug allergy since avoidance has been deemed protective. Recent data have shown that an inaccurate penicillin allergic label incurs increased hospital stays, the use of more expensive alternative antibiotics, more complications, delays in treatment time, and antibiotic resistance [12]. Strong evidence indicates that an inaccurate drug allergy label leads to increased health risks, and in patients with mastocytosis, it can lead to unnecessary avoidance of medications for which the risk of MC activation may be low and may be modified with appropriate diagnosis and premedications [13,14,15,16]. We provide updated information on current approaches to drug allergy and Hymenoptera venom hypersensitivity diagnosis and management in mastocytosis.

## 2. Nonsteroidal Anti-Inflammatory Drugs (NSAIDs)

Nonsteroidal anti-inflammatory drug (NSAID) hypersensitivity is one of the leading causes of drug hypersensitivity reactions (DHRs) worldwide, accounting for up to 25% of all DHRs and affecting 0.3–2.5% of the general population [17,18]. Patients with mastocytosis have a higher prevalence of adverse reactions than the general population; prior reports indicate up to 14% of patients may have had an adverse NSAID reaction [19,20]. Concern for severe DHRs including anaphylaxis has led to frequent avoidance of NSAIDs in patients with mastocytosis [21,22,23,24].

Emerging data suggests that NSAID-induced adverse reactions and anaphylaxis may be less frequent than previously suggested [20,23,25]. One retrospective study examining 388 patients with mastocytosis found that 11.3% of patients reported an adverse reaction to NSAIDs, the majority (89%) isolated to the skin [22]. Symptoms included flaring of cutaneous lesions (89%), flushing (26%), angioedema (21%), pruritus (21%), and urticaria (16%), and only 2.8% of patients had confirmed NSAID-induced anaphylaxis. Another study of 681 cutaneous and systemic mastocytosis patients from Spain reported that 87% of adults and 91% of children tolerated NSAIDs [21]. Of the patients with adverse reactions, 60% reacted to a single drug, while 40% had reactions to multiple NSAIDs and the most frequent culprits were aspirin (ASA) (17%), metamizole (12%), or coxibs (12%) [21]. All pediatric patients tolerated paracetamol, supporting its use as a suitable alternative to those with NSAID intolerance. The statistics from these studies suggest that the true prevalence of DHRs due to NSAID use among mastocytosis patients is less than previously assumed.

ASA irreversibly inhibits cyclo-oxygenase 1, leading to decreased prostaglandins production, and its use is important in mastocytosis as it may improve symptoms induced by excessive production of prostaglandins such as flushing [9,20,26]. A double-blind, placebo-controlled study of 50 patients with mastocytosis found that 98% of patients tolerated an ASA challenge [24] and a retrospective chart review of 191 patients from the same group found that 95.9% tolerated ASA, verifying the high tolerance of ASA [24]. 

Predicting NSAID hypersensitivity may be challenging, and several studies have aimed to identify risk factors. Female sex, advanced age, history of drug-induced anaphylaxis, higher baseline tryptase levels, and multilineage hematopoietic involvement by the KIT D816V mutation have been associated with a higher probability of NSAID reactions [21]. Rama et al. utilized this information to develop a risk stratification model to assess those at higher risk of adverse NSAID reaction [21]. Although the model has low positive predictive value, it identifies patients at low risk for reactions and can be used as screening tool. Further research is needed to improve upon this risk stratification model and make it an effective tool for identifying patients at high risk for DHRs. While patients who tolerated NSAIDS before the diagnosis of mastocytosis should continue their use without restriction, a challenge based on risk stratification should be used in patients without known tolerance recommendations for NSAIDS [21,22,23,24,27]. For patients with a low DHR score, an initial challenge to the desired NSAIDs should be performed in a supervised setting. For high-risk patients, a supervised challenge to a selective COX-2 inhibitor (such as celecoxib) could be considered [24]. In the case of NSAID intolerance, most patients with mastocytosis will tolerate paracetamol or acetaminophen as an alternative to NSAIDs [21]. 

## 3. Antibiotics 

Patients with systemic mastocytosis have a higher risk of antibiotic-induced hypersensitivity than the general population [19]. Adverse reactions have been reported to multiple antibiotic classes including beta-lactams, fluoroquinolones, and vancomycin [28,29,30,31].

Beta-lactam antibiotics are frequently identified as culprits in the literature, but there are no current studies which have assessed the prevalence and incidence among mastocytosis patients. One retrospective study specifically looked at the rate of antibiotic reactions in adult patients with mastocytosis. In the cohort of 239 patients, 176 patients met WHO criteria for systemic mastocytosis, 18 had mastocytosis in the skin (MIS), and 45 patients had monoclonal mast cell activation syndrome (MMAS). The study found that 34 patients (14.2%) had an allergic reaction to antibiotics and 2 presented with anaphylaxis (0.8%). Symptoms were triggered by a beta-lactam in 63% of the cases [32]. Additionally, in a cohort of 133 children with cutaneous mastocytosis, 6 (4.5%) had a reaction to β-lactam antibiotics [33]. More studies are needed to define the frequency of allergic reactions to antibiotics in the different subtypes of mastocytosis, such as cutaneous versus systemic disease, and among patients of different age demographics (pediatric and adult). 

Acute hypersensitivity reactions to drugs are thought to occur via both IgE- and non-IgE-dependent pathways [34]. In the case of quinolones, a report indicated that 54.5% of reactive patients had specific IgE to a quinolone [35], but other studies have failed to confirm these results and implicated non-IgE pathways such as MRGPRX2 [30,36]. MRGPRX2 is a novel G-protein coupled receptor expressed on cutaneous mast cells, and its expression is increased in patients with urticaria and cutaneous mastocytosis [37]. MRGPRX2 induces non-IgE activation of mast cells when binding to drugs with THIQ motifs such as neuromuscular blockers, quinolones, and basic compounds, such as vancomycin [31]. A case report describes a 46-year-old female with systemic mastocytosis without cutaneous involvement that had two anaphylactic episodes triggered by ciprofloxacin. Additionally, this patient reported three severe reactions to Hymenoptera venom. The patient had negative skin testing to both Hymenoptera venom and ciprofloxacin, supporting that these reactions may have been mediated by the MRGPRX2 receptor [38].

Because there are currently no biomarkers to assess the expression of MRGPRX2 in SM patients, THIQ-motif containing general anesthetics, quinolones, and vancomycin should be avoided when possible due to the potential for mast cell activation. Patients can take these antibiotics if previously tolerated, but a supervised challenge is recommended if tolerance is unknown. All patients with a history of antibiotic hypersensitivity reactions should be assessed with skin testing and have a supervised challenge if negative prior to antibiotic re-exposure. Patients with low risk should be evaluated by an allergist for drug allergy de-labelling and patients with positive skin test and/or positive challenge should be labeled as allergic and desensitization is recommended if the drug is needed as first-line therapy and there are no effective alternatives [13,15,28].

In general, patients with mastocytosis would benefit from additional drug hypersensitivity testing modalities. Skin testing is generally preferred for drugs with high sensitivity/specificity (such as penicillin) [39]. Serum-specific IgE has also been used, classically for penicillin allergy. Finally, basophil activation testing (BAT) may be helpful for some medications, particularly to risk-stratify patients prior to provocation challenges [40]. 

Vancomycin and fluoroquinolones may induce reactions through non-IgE mechanisms, potentially MRGPRX2, and their use should be limited in patients with mastocytosis.

Patients for which these drugs cannot be substituted may benefit from a slow infusion for vancomycin and supervised challenge with or without pre-medication for quinolones [38,41]. Other antibiotics do not require supervised challenges for introduction. 

## 4. Vaccines

While some triggers for mast cell activation and anaphylaxis in mastocytosis patients are well characterized, the impact of vaccination on mastocytosis patients is poorly understood. Among the general United States population, the incidence of adverse reactions to vaccination is between 3 and 6%; few of these reactions have the potential to be life-threatening [23]. It is estimated that 1 in 1 million vaccine administrations result in anaphylaxis. 

Most vaccine studies among mastocytosis patients have been conducted via retrospective chart review and have concluded that patients with mastocytosis may have a higher incidence of adverse reactions compared to the general population [42,43]. Verona University Hospital conducted a study of 72 adults and children with mastocytosis, receiving 137 vaccines total, and found that vaccines were well tolerated without reaction, particularly in adult patients [42]. None of the adults (0/57 vaccines) and seven of the children (7/80 vaccinations) experienced an adverse event to a vaccine administration. No events of anaphylaxis were reported. Of note, 57% of adverse vaccine events were related to the hexavalent (diphtheria, tetanus, pertussis, poliovirus, H. Influenzae type b, and hepatitis B) vaccine [43]. A report of a severe systemic reaction with whole body blistering in a child with DCM who received five vaccines promoted caution of multiple vaccinations in children with extensive cutaneous mastocytosis [44]. Further research must be conducted to better characterize the relationship between age, type of mastocytosis (i.e., systemic, diffuse cutaneous, etcetera), and risk for adverse reaction to vaccination.

Much of the research on vaccines in adults with mastocytosis has sprung from the need to immunize patients against COVID-19. The initial observation of two patients with ISM who received mRNA vaccination safely with antihistamine premedication provided evidence of the lack of severe mast cell activation symptoms or anaphylaxis [45]. With premedication, 2 of 73 (2.3%) patients with mastocytosis developed symptoms to the Pfizer mRNA vaccines. This is comparable to 2% of atopic patients without a mast cell disorder [46]. The largest study of COVID-19 vaccination administration among individuals with mastocytosis included 323 patients, receiving a total of 666 vaccines. It was found that 6% of vaccinations resulted in adverse events, with cutaneous symptoms being the most common. The anaphylaxis rate was higher among mastocytosis patients than in the general population, prompting recommendations for pre-medication and an increased observation period. It is also recommended that patients carry epinephrine autoinjectors at the time of vaccination [43].

An Italian study comparing vaccination safety among children with mastocytosis and their siblings (without mastocytosis) found that the rate of reaction to vaccinations in children with mastocytosis (4 per 634 vaccines) was higher than the rate of reaction of 2.3 cases per 10,000 doses reported for the general population [47]. The four reactions were mild, transient, and resolved upon treatment with antihistamines. The National Institutes of Health conducted a retrospective chart review of 94 children with mastocytosis who received 2136 vaccines, finding that 4% of children had urticaria, anaphylaxis, and exacerbated skin lesions [23]. Case studies have reported children developing bullous skin eruptions and a mastocytoma at the site of vaccination [44,48,49]. The frequency and severity of mast cell activation and adverse reactions to vaccines is higher in pediatric mastocytosis patients with greater skin involvement, diffuse cutaneous mastocytosis, and elevated basal serum tryptase [44,50]. Premedication and post-injection supervision for 30 min may reduce symptoms and ensure all children with mastocytosis follow recommended immunization guidelines. 

Understanding the mechanism of allergic and adverse reactions of vaccines is important to address the declining rates of immunization compliance and rise in uncertainty or fear of vaccination [51]. In patients with mastocytosis, vaccine adverse reactions are thought to be related to nonspecific immune activation and to multiple simultaneous vaccines. As reported by Zanoni et al. [42], of four children who presented mast cell activation symptoms following hexavalent vaccine, two were subsequently able to tolerate the individual vaccine components and the other two received several components without adverse reaction.

All age-appropriate vaccines should be administered to children with mastocytosis. For those with diffuse cutaneous mastocytosis or high risk of anaphylaxis, premedication and limiting administration to a single vaccine at a time may reduce the risk of adverse effects [3,45,52].

## 5. Perioperative Medications and Anesthesia

Patients with mastocytosis are at higher risk for perioperative hypersensitivity reactions [53]. Antibiotics are one of the most common causes of perioperative anaphylaxis and are addressed in the ‘antibiotic’ section above. Several studies have examined the use of general anesthesia and associated medications in patients with mastocytosis. A retrospective study of 501 patients with mastocytosis identified that 2% of adults had a DHR and 0.4% of adults experienced anaphylaxis. In its pediatric counterpart, 4% developed a hypersensitivity reaction to anesthesia, and 2% had anaphylaxis [54]. While these percentages indicate a statistically significant risk of anaphylaxis, it is also known that the same anesthesia is generally tolerated if administered under premedication [55]. A similar albeit smaller study on pediatric populations (*n* = 22) identified about 9% having a DHR to general anesthesia [56]. A detailed history including any past medication reactions should be obtained prior to surgery to evaluate risk and the need for pre-medications. 

Adverse drug reactions in the perioperative setting may be caused by a variety of agents, although neuromuscular blocking agents (NMBAs) and opiates are the most reported [53,57]. These can be categorized into three pharmaceutical classes that have differing potentials for adverse reactions. Non-depolarizing NMBAs such as atracurium, rapacuronium, and mivacurium have been associated with higher DHRs and are not recommended for perioperative use for mastocytosis patients [58,59,60]. It is thought that most adverse reactions to NMBAs are mediated via the MRGPRX2 receptor in a concentration dependent manner [34,36,37]. 

In contrast to nondepolarizing NMDAs, Succinylcholine does not interact with MRGPRX2 and is considered safe for patients with mastocytosis [60,61,62]. Sugammadex can safely be used to reverse the neuromuscular blockade in patients with systemic mastocytosis [63].

Opiates vary in their ability to directly activate mast cells. Generally, natural opiates can cause mast cell degranulation in a dose-dependent manner [64]. This is observed with morphine, codeine, meperidine/pethidine, and buprenorphine [65], while synthetic opiates do not elicit similar histamine and tryptase release. Piperidine-derived opiates (fentanyl, sufentanil, alfentanil, naloxone, etc.) show a low degree of mast cell activation and little release of tryptase and histamine [59] and are considered safer for mastocytosis patients. Other synthetic opiates (oxycodone, hydromorphone, tramadol) have similar low histamine response in vivo [66]. There are no studies which have addressed the incidence and prevalence of reactions to natural versus synthetic opiates in mastocytosis, and while synthetic opiates are preferred, other analgesics should be used first due to opiate addiction potential. Acetaminophen and most NSAIDS are generally well tolerated in mastocytosis patients as described above [21].

Local anesthetics are generally considered to be safe for patients with mastocytosis. Despite limited data investigating the prevalence of reactions in mastocytosis patient populations, a retrospective study of the Spanish network on mastocytosis, REMA, found that out of 45 pregnant ISM patients, 35 were given an epidural or local anesthesia, and only 5 cases displayed symptoms of mast cell mediator release. Symptoms were limited to pruritus, generalized erythema, and flushing [54]. Another retrospective study compared local anesthetic allergies between the general population and mastocytosis patients. Out of a group of 252 patients with ISM, only 5 have a history of local anesthetic allergy, either to lidocaine or mepivacaine [67].

Evaluation of perioperative anaphylaxis in this patient population is challenging. We recommend full evaluation for any patients with mast cell mediator symptoms during the peri-operative period. In high-risk patients, careful evaluation prior to invasive procedures should be considered. The goal of evaluation is to identify a culprit drug and/or make recommendations to ensure safety during future surgery. Many common triggers may be evaluated via skin testing [53,68] and standardized skin testing protocols are available for common drugs [69]. 

In many situations, no culprit will be identified. In these cases, we suggest empiric premedication with H_1_-antihistamines and systemic corticosteroids. Potential culprit medications should be identified and switched to an alternative medication if feasible [27,67]. Premedication may also be helpful for higher-risk patients with a history of severe anaphylaxis, high mast cell burden, or past perioperative reactions. Finally, it is important to acknowledge that premedication has not been studied in a randomized, controlled trial. Therefore, careful discussion of risks and benefits must be assessed for each individual patient [53]. 

## 6. Radiocontrast Media

Adverse reactions to radiocontrast media (RCM) are uncommon and occur in less than 1% of the general population [70]. There are limited epidemiologic data in patients with mastocytosis, although case reports indicate the potential for MC activation symptoms including anaphylaxis [23,64,71]. Brockow et al. reported 120 patients with anaphylaxis and found that RCM was a trigger in 2 of 120 reported reactions [64]. In another study of 220 patients with RCM hypersensitivity, no patients were found to have a clonal mast cell disorder [72]. 

There are limited data to guide workup in patients with RCM hypersensitivity. Skin testing may predict the likelihood of recurrent anaphylaxis [73]. In addition, skin testing to alternative contrast agents may be helpful to identify alternative agents better tolerated by the patient [74]. Unfortunately, there are no studies that have examined the severity of adverse reactions, workup, or alternative therapies, specifically in patients with mastocytosis. 

Although there are no controlled studies supporting the use of premedication, the benefits likely outweigh the risks. All patients who report prior reactions should receive premedication for future contrast administration. Nonsedating second-generation antihistamine and systemic steroids are recommended by consensus work groups [23]. 

## 7. Venom Hypersensitivity

Patients with systemic mastocytosis are more likely than the general population to experience anaphylaxis following Hymenoptera sting [68,75]. Anaphylaxis is reported in approximately 0.3–8.9% in the general population comparted to 20–30% in mastocytosis patients [23,76,77,78]. Hymenoptera anaphylaxis is the most common cause of anaphylaxis in patients with clonal mast cell disorders and is responsible for a disproportionate share of severe anaphylaxis, particularly episodes involving hypotensive syncope requiring resuscitation and several doses of epinephrine. 

Any patients with mastocytosis may suffer anaphylaxis from Hymenoptera venom. However, a subset of male patients with isolated bone marrow mastocytosis (absence of cutaneous lesions and low baseline tryptase) are often at the highest risk [79,80,81]. Cardiovascular collapse with hypotensive syncope in the absence of urticaria is typically the presenting event, prompting the evaluation of serum tryptase and further studies leading to the diagnosis of mastocytosis [82]. The diagnosis of mastocytosis in patients with severe Hymenoptera anaphylaxis is critical to promote lifesaving interventions such as venom immunotherapy (VIT) and/or the use of recently approved TKIs. 

The diagnosis of venom hypersensitivity is confirmed via serum-specific IgE to Hymenoptera venom and/or with skin testing [75,83]. The authors suggest initial evaluation with serum-specific IgE followed by skin testing if there is no serologic evidence of sensitization. We suggest starting with prick testing at 100 mcg/mL concentration. If negative, intradermal testing is conducted at 0.01 mcg/mL, 0.1 mcg/mL, and 1 mcg/mL [75,84]. A positive skin test at any level indicates sensitization. The potential for adverse outcomes is high with venom hypersensitivity, and it is thus imperative to identify venom sensitization when present [85].

Multiple reports have described the efficacy of VIT in mastocytosis [85,86]. Although clearly beneficial, VIT may also be complicated by frequent adverse reactions, particularly during the build-up phase. More than 20% of patients with mastocytosis report side effects from VIT [78,83]. Adverse reactions to VIT may be mitigated by protocol adjustments such as rush or ultrarush therapy [87,88,89]. Several publications support the use of omalizumab as an premedication to reduce severity of adverse reactions to venom [90,91,92,93]. A successful 6 h ultrarush venom desensitization protocol by means of 12 incremental injections was recently developed for use in patients with Hymenoptera anaphylaxis and clonal mast cell disorders [93]. The protocol involves premedication with omalizumab for patients with severe anaphylaxis followed by venom injections every 30 min. This achieves maintenance doses in ~6 h. This protocol is most useful in high-risk patients such as beekeepers, golfers, or those who may encounter Hymenoptera while working. The Hymenoptera ultrarush desensitization protocol is shown in Table 1. Ultrarush therapy has been used in many other settings without omalizumab, although none of these protocols involve patients with clonal mast cell disorders [88,94,95].

In patients with clonal mast cell disorders and venom anaphylaxis, venom immunotherapy is recommended for life. Two case reports describe fatal stings in patients who were previously on VIT and discontinued, indicating the lifesaving nature of VIT in this patient population [85].

## 8. Drug Hypersensitivity in Children and Pregnant Women

Children are most frequently diagnosed with cutaneous mastocytosis, a variant that is associated with less anaphylaxis than systemic mastocytosis [64]. Children with cutaneous mastocytosis are often advised to avoid a variety of medications including NSAIDs, antibiotics, RCM, and vaccines. There are few case reports describing adverse reactions in children with mastocytosis although antibiotics and vaccines have both been reported [44,48,96]. Empiric avoidance deprives children of essential antibiotics, vaccines and pain medications, and all children should undergo evaluation at the time of adverse reactions. 

Pregnancy has a mixed effect on mast cell activation symptoms. Matito et al. reported forty-five pregnancies in 30 women with mastocytosis [97]. In 37 cases, patients were exposed to epidural, local, or general anesthesia. Symptoms were reported in 5/37 (13%) of cases; no patient had anaphylaxis. Pregnancy does not appear to be associated with increased mast cell activation and routine anesthetic procedure should be used. Premedication may be helpful in patients with a history of drug-induced prior mast cell activation symptoms or anaphylaxis. It is also important to note that skin testing and provocation procedures should generally be avoided in pregnancy unless there are no reasonable alternatives. 

## 9. Hereditary Alpha-Tryptasemia

Hereditary alpha-tryptasemia (HαT) is a genetic trait characterized by extra-allelic copies of the alpha tryptase gene at TPSAB2 [98]. Several studies have estimated a prevalence of 5–7% in the general population. The prevalence of HαT is higher (12–18%) in systemic mastocytosis [99,100]. This genetic trait modifies the severity of anaphylaxis and allergic disorders, which has implications for conditions such as Hymenoptera venom-triggered anaphylaxis, idiopathic anaphylaxis, and food allergy. 

While the full clinical phenotype is under investigation, existing data supports a strong connection with severe anaphylaxis. Individuals with HαT are at higher risk for grade III/IV anaphylaxis, particularly involving cardiovascular compromise. This has been reported in several unselected cohorts [100,101]. 

The risk of drug allergy in hereditary alpha tryptasemia has not been specifically determined [100,102,103], and evaluation for TPSAB1 tryptase gene duplication in patients with systemic mastocytosis and severe anaphylaxis can help address risk when exposed to Hymenoptera venoms, foods, and drugs. 

## 10. Conclusions

Clonal mast cell activation disorders are associated with an increased risk of adverse reactions to multiple drugs including NSAIDs, antibiotics, vaccines, radiocontrast media, perioperative medications, and Hymenoptera venom. The frequency of adverse reaction for each drug class is displayed in Table 2. While patients may have higher expression of MRGPRX receptors in skin mast cells, the duplication of TPSAB1 genes which are found in higher frequency than in the general population with high tryptase levels, may impact these reactions. It is important to identify and accurately diagnose drug hypersensitivity to reduce the risk of adverse drug reactions and anaphylaxis. Pre-medications are recommended for high-risk patients.

For patients with a history of adverse reactions, testing is recommended to avoid mislabeling. Many patients may be inappropriately labeled ‘allergic’ without a history of reaction or positive testing, and de-labeling is strongly recommended to prevent unnecessary avoidance to first-line medications. Objective testing and challenges will increase the repertoire of available medications. 

## Figures and Tables

**Table 1 diagnostics-14-00123-t001:** Hymenoptera venom ultrarush protocol. Adopted with permission from Giannetti et al., 2020 [93].

Step	Time (min)	Dilution	Concentration (µg/mL )	Volume (mL) Subcutaneously	Dose (µg)	Cumulative Dose (µg)
1	0	1:1000	0.3	0.2	0.06	0.1
2	30	1:1000	0.3	0.4	0.12	0.2
3	60	1:1000	0.3	0.8	0.24	0.4
4	90	1:100	3	0.2	0.6	1.0
5	120	1:100	3	0.4	1.2	2.2
6	150	1:100	3	0.8	2.4	4.6
7	180	1:10	30	0.2	6	10.6
8	210	1:10	30	0.4	12	22.6
9	240	1:10	30	0.8	24	46.6
10	270	1:1	300	0.1	30	76.6
11	300	1:1	300	0.25	75	151.6
12	330	1:1	300	0.5	150	301.6

Three dilutions of standard venom extract are created (1:1000, 1:100, 1:10). Each dose is administered in 30-min intervals. The first maintenance dose was administered 2 wk after desensitization, followed by once-monthly injections.

**Table 2 diagnostics-14-00123-t002:** High-risk medications in mastocytosis and estimated prevalence.

	Prevalence of Adverse Reactions in Adults (%)	Prevalence of Adverse Reactions in Children (%)	References
NSAIDs	10–13%	<9%	[17,19,20]
Antibiotics	<14%	Limited data	[18,22]
Vaccines	2–6%	0–9%	[28,39,40]
Perioperative medications	2%	4–9%	[50,51]
Radiocontrast Media	Rare	Unknown	[59,66]
Hymenoptera allergy	~20%	Unknown	[27,53,69]

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
