# Peer review of "Challenges in Drug and Hymenoptera Venom Hypersensitivity Diagnosis and Management in Mastocytosis"

_diagnostics, 2024, doi:10.3390/diagnostics14020123_

Round 1
Reviewer 1 Report
Comments and Suggestions for Authors
This is interesting and practical article on Drug/hymenoptera hypersensitivity in mastocytosis. I believe its interesting for physicians and fights common myths in mastocytosis and drugs.
In general the article is ok, however the text still requires some polishing and expanding some parts. The comments are below.
Part 2 NSAIDS. Authors might consider in vitro studies short description such as Basophil Activation Test.
Part 3.
Line 129 Skin testing in mastocytosis may come false negative due to probably absorption of IgE on mast cells. Some scientists suggest that lower cut-off point of serum IgE should be used in SM. Thus negative tests do not exclude IgE mechanism. Also BAT might be an option in some drugs.
Line 138 Please add at the end of sentence "with no effective alternatives"
Part 4. - no comments
Part 5. Requires expansion. Perioperatively also antibiotics and disinfectants (chlorhexidinemost prevalently) can cause DHR. Antibiotics are described above so you can just add them to sentence line 134. This part also misses the allergy workup possibilities as skin testing is done with most of these drugs and with some of them (local anesthetics) provocations can be done. With some drugs such as morphine the tests come often false positive due to probably direct mast cell activation.
Local anesthetic allergy is exceptionally rare even in patients with mastocytosis!
Part 6. Also please add possible diagnostic skin testing and provocations
Part 7. Line 273 Also in patients with mastocytosis Immunotherapy should be lifelong. Please add short description of diagnostic procedures: skin tests, specific IgE
Line 280-287 Ultrarush is a standard protocol used for example in Poland and many places in Europe for all patients (though protocols vary). Some of them do not require omalizumab and are actually very safe.
Part 8 Line 296-298 I disagree with this sentence . It suggests that we have to do supervised challenges in all medications in children with mastocytosis.
In this part it is worth mentioning the limitations of skin testing/drug provocations in pregnancy.
Author Response
This is interesting and practical article on Drug/hymenoptera hypersensitivity in mastocytosis. I believe its interesting for physicians and fights common myths in mastocytosis and drugs.
In general the article is ok, however the text still requires some polishing and expanding some parts. The comments are below.
Thank you for your time and effort reviewing this manuscript. We have added point-by-point responses in red below.
Part 2 NSAIDS. Authors might consider in vitro studies short description such as Basophil Activation Test.
There are no validated in vitro or skin testing procedures in the general population to confirm NSAID hypersensitivity. The discussion of the risks/benefits of experimental procedures such as BAT for NSAID allergy is outside the scope of this review.
Part 3.
Line 129 Skin testing in mastocytosis may come false negative due to probably absorption of IgE on mast cells. Some scientists suggest that lower cut-off point of serum IgE should be used in SM. Thus negative tests do not exclude IgE mechanism. Also BAT might be an option in some drugs.
Agree. Two sentences regarding skin testing, sIgE, and BAT were added to this section.
Line 138 Please add at the end of sentence "with no effective alternatives"
Thank you, this has been added.
Part 4. - no comments
Part 5. Requires expansion. Perioperatively also antibiotics and disinfectants (chlorhexidinemost prevalently) can cause DHR. Antibiotics are described above so you can just add them to sentence line 134.
Added a sentence to the beginning of the perioperative medications section to highlight that antibiotics are covered in the ‘antibiotic’ section earlier in the paper.
This part also misses the allergy workup possibilities as skin testing is done with most of these drugs and with some of them (local anesthetics) provocations can be done. With some drugs such as morphine the tests come often false positive due to probably direct mast cell activation.
A paragraph describing skin testing and premedication has been added to the manuscript.
Local anesthetic allergy is exceptionally rare even in patients with mastocytosis!
Agree! This section was not altered per the editors request for total word count >4000
Part 6. Also please add possible diagnostic skin testing and provocations
Several sentences were added regarding skin testing and provocation challenges to RCM.
Part 7. Line 273 Also in patients with mastocytosis Immunotherapy should be lifelong. Please add short description of diagnostic procedures: skin tests, specific IgE
Lifelong therapy is important. This has been added. Short description of workup modality is in the text with reference to AAAAI practice parameters for specific diluations, etc.
Line 280-287 Ultrarush is a standard protocol used for example in Poland and many places in Europe for all patients (though protocols vary). Some of them do not require omalizumab and are actually very safe.
Also agree with this. A sentence has been added to highlight that omalizumab is not needed for all ultrarush protocols and is specific to those with mastocytosis.
Part 8 Line 296-298 I disagree with this sentence . It suggests that we have to do supervised challenges in all medications in children with mastocytosis.
Agreed. That was not the intent of the sentence. It has been removed.
In this part it is worth mentioning the limitations of skin testing/drug provocations in pregnancy.
Agree and a sentence has been added to highlight this.
Reviewer 2 Report
Comments and Suggestions for Authors
The review article provides an overview of common agents causing adverse drug reactions in mastocytosis patients and also review diagnososis and management of drug hypersensitivity and Hymenoptera venom allergy in this patients.
My comments:
1. This review would benefit from clarification about the relationship between the KITD816V mutation and NSAID reactions. Are there any report indicating that these reactions more common in patients with this mutation?
2. Information on the recommended approach for antibiotic administration in mastocytosis patients is lacking.
Author Response
The review article provides an overview of common agents causing adverse drug reactions in mastocytosis patients and also review diagnosis and management of drug hypersensitivity and Hymenoptera venom allergy in this patients.
Thank you for the time and effort reviewing our manuscript. We have provided a point-by-point response below.
My comments:
- This review would benefit from clarification about the relationship between the KITD816V mutation and NSAID reactions. Are there any report indicating that these reactions more common in patients with this mutation?
Unfortunately, there are no data directly examining adverse reactions to NSAIDs between patients with KIT D816V+ SM and those with KIT D816V- SM.
- Information on the recommended approach for antibiotic administration in mastocytosis patients is lacking.
We have added another paragraph describing our approach to antibiotics in this patient population.
Round 2
Reviewer 1 Report
Comments and Suggestions for Authors
The authors adressed all issues. I have no further comments. Good job.